# Accelerometer-Derived Intensity Thresholds Are Equivalent to Standard Ventilatory Thresholds in Incremental Running Exercise

**DOI:** 10.3390/sports11090171

**Published:** 2023-09-05

**Authors:** Matthias Schützenhöfer, Philipp Birnbaumer, Peter Hofmann

**Affiliations:** Exercise Physiology, Training and Training Therapy Research Group, Institute of Human Movement Science, Sport and Health, University of Graz, 8010 Graz, Austria; matthias.schuetzenhoefer@outlook.com (M.S.); philipp.birnbaumer@uni-graz.at (P.B.)

**Keywords:** exercise prescription, activity measures, thresholds

## Abstract

Accelerometer cut-points are commonly used to prescribe the amount of physical activity, but this approach includes no individual performance measures. As running kinetics change with intensity, acceleration measurements may provide more individual information. Therefore, the aim was to determine two intensity thresholds from accelerometer measures. A total of 33 participants performed a maximal incremental running test with spirometric and acceleration (Axivity AX3) measures at the left and right tibia. Ventilatory equivalents (VE/VO_2_, VE/VCO_2_) were used to determine a first and second ventilatory threshold (VT_1_/VT_2_). A first and second accelerometer threshold (ACT_1_/ACT_2_) were determined within the same regions of interest from vector magnitude (|v| = √(ax^2^ + ay^2^ + az^2^). Accelerometer data from the tibia presented a three-phase increase with increasing speed. Speed at VT_1_/VT_2_ (7.82 ± 0.39/10.91 ± 0.87 km/h) was slightly but significantly lower compared to the speed at ACT_1_/ACT_2_ from the left (7.71 ± 0.35/10.62 ± 0.72 km/h) and right leg (7.79 ± 0.33/10.74 ± 0.77 km/h). Correlation analysis revealed a strong relationship between speed at thresholds determined from spriometric data or accelerations (r = 0.98; *p* < 0.001). It is therefore possible to determine accelerometer thresholds from tibia placement during a maximal incremental running test comparable to standard ventilatory thresholds.

## 1. Introduction

Performance diagnostics are a fundamental tool in health and disease [1] but are usually time consuming and expensive. Therefore, easy-to-apply techniques leading to reliable and valid results are in demand [2]. Cardiopulmonary Exercise Testing (CPET) as the “gold standard” of performance diagnostics allows the differentiation of performance into distinct metabolic and cardiorespiratory zones, which are separated by two thresholds or turn points being a state-of-the-art accepted model [1,3,4]. Systemic variables such as ventilation (VE), heart rate (HR), and blood lactate concentration (La) are frequently used to obtain thresholds [3]. Also, to assess submaximal and maximal cardiorespiratory fitness, these thresholds, as well as maximal values, such as power output (P_max_) or oxygen uptake (VO_2max_), are commonly determined using incremental cycle ergometer [5,6,7] or running exercise [8,9,10]. A threshold-based prescription of exercise intensity provides valuable information for healthy subjects, athletes, or patients in order to optimize the training process and to provoke intensity- and volume-specific effects on performance [11], prevention [12], or chronic diseases [13]. However, these methods are limited to a small number of specific groups and rarely applicable to outdoor running as well as to the public.

On the other hand, easy and cheap non-invasive tools such as accelerometers are frequently used in everyday life to quantitatively measure activity as well as to estimate and prescribe exercise intensity [14]. Usually, intensity prescription with accelerometer cut-points refers to fixed absolute intensity classes expressed in multiples of the metabolic equivalent (MET) [15,16]. One major limit of such measures is that they solely obtain the absolute amount of activity without relation to individual intensity thresholds or limits. Although this approach is widely applied, it was also critically discussed as no individual performance measures are included [17].

However, accelerometer data are suggested to provide more individualized information. It was already shown that tibial acceleration is modified by running speed [18] and changed with increasing duration during running with constant speed [19]. Faster running speeds were associated with increased tibial acceleration, irrespective of running surface, footwear, running experience, or whether the velocity was fixed or self-selected [18]. This relationship was suggested to be linear (but lacking statistical analyses) from a study where 10 well-trained runners increased their velocity from 3.4 to 4.5 m/s. They showed, an average increase in tibial acceleration by 34% with every 1 m/s increase [20]. Sheerin et al. [21] showed similar results with an average 38% increase in tibial acceleration when speed was increased in three steps by 1 m/s from 2.7 to 3.7 m/s (9.7 to 13.3 km/h). However, running velocity and acceleration correlated only moderately, and only 19% of tibial acceleration could be explained by velocity. They supposed that the relationship might be non-linear at slower, and potentially faster, running velocities and that acceleration measures probably show inflection points at different velocities. Furthermore, they suggested that other factors like stride-length (SL) and -frequency (SF) substantially account for the changes in acceleration. In a systematic review, Apte et al. [22] already showed that exercise intensity in running significantly influenced biomechanical parameters such as SF and SL, which changed systematically with increasing running speed [23,24]. During running at submaximal intensities, experienced runners were shown to adopt an optimal combination of SL and SF that minimized metabolic cost, which could not be maintained during high intensity running [25,26]. Furthermore, with increasing speed, SL and SF were shown to increase in different ratios, where SL increased by 15% more than SF when speed was increased from 60% to 80% of the maximal velocity in an incremental running test. However, the increase in speed from 80% to 100% of the maximal velocity only caused a 3.5% higher increase in SL compared to the increase in SF [27]. Increasing running speed in small increments could therefore allow the detection of systematic changes in the time course of tibial acceleration as a result of increasing running speed and the concomitant non-linear adaptation of both SL and SF.

Contrary to an expected linear time course [20,21], measures within a recent study focusing on changes of tibial acceleration to classify soccer-specific activities [28] showed a non-linear increase in accelerations in a maximal incremental running test, which was similar to the three-phase increase in ventilation as also supposed by Sheerin et al. [21]. Therefore, three-dimensional acceleration measures from the tibia are suggested to distinguish between different phases of metabolism similar to standard physiological measures. Such a novel analysis would allow the obtainment of not only quantitative information from accelerometers but also the quality of individual activity in relation to individual metabolic and cardio-respiratory target zone thresholds, respectively. Easily accessible smart phone applications enable a widespread population-based use [29]. Therefore, the aim of this study was to determine a first and a second intensity threshold according to a three-phase model of exercise intensity from accelerometer measures on the tibia. We hypothesized that the time course of accelerometer measures from the distal part of the tibia is comparable to physiological variables and allows the determination of two thresholds in all subjects that are significantly related and not significantly different from the standard first (VT_1_) and second (VT_2_) ventilatory thresholds. Additionally, we assumed that left and right leg measures are similar and that no differences can be found with respect to sex.

## 2. Materials and Methods

### 2.1. Participants

In total, a group of 33 participants (12 women; 23.9 ± 2.6 yrs, 177.7 ± 8.0 cm, 71.2 ± 9.4 kg,) were tested at the local university outdoor running track. All participants were trained, highly active, and healthy students familiar with incremental running tests. Before the start of the study, all participants received detailed information regarding the testing protocol and measures, and they signed a written consent form. The study protocol was conducted according to the Declaration of Helsinki and approved by the local ethics committee (GZ. 39/45/63 ex 2020/21).

### 2.2. Protocol

Participants performed one maximal incremental running test on a standard 400 m outdoor running track. The incremental test started at 6 km/h, and running speed was increased by 0.5 km/h every 100 m up to the individual maximum running speed according to the adapted protocol prescribed by Conconi et al. [30]. To control speed, markers were placed every 20 m on the running track, and participants were paced by audio signals from a computer-based pacer. Participants were urged to pass the markers with the pacer signal to keep the given pace until subjective exhaustion. When participants were no longer able to follow the given pace (be at the marker with the signal), the test was stopped. A three-minute period of upright standing was performed before and after the test to measure rest and recovery kinetics. VO_2max_ was determined as the mean value of oxygen uptake during the last thirty seconds of exercise. VO_2max_ was achieved if either the respiratory exchange ratio (RER) was greater than 1.1 or the individual age predicted heart rate (220-age) was present.

### 2.3. Measurements

During the test, gas exchange data were measured continuously in breath-by-breath mode using a portable gas analyzer (CORTEX METAMAX 3B, Cortex Biophysik GmbH, Leipzig, Germany). Calibration of ventilation, as well as O_2_ and CO_2_ gas sensors, was performed prior to each test according to the manufacture’s guidelines. Heart rate was continuously measured with a chest belt and a heart rate monitor (Polar S810i; Polar Electro, Kempele, Finland), and data were stored in 5 s intervals for further analyses. All participants were equipped with two accelerometers (Axivity AX3, Axivity Ltd., Newcastle upon Tyne, UK) for activity measurements. The accelerometers were attached once on the left and once on the right tibia midpoint between kneecap and ankle. For all accelerometer measures, a data sampling frequency of 100 Hz and a sampling range of ±16 g was initialized.

### 2.4. Determination of Thresholds

Gas exchange data and raw triaxial accelerations were transferred into Microsoft Excel files (Microsoft Corporation, Redmond, WA, USA) in 5 s epochs using the manufacturer’s software. From raw triaxial accelerations the vector magnitude (|v|) was calculated using |v| = √(ax^2^ + ay^2^ + az^2^) (m/s^2^), and physiological and accelerometer thresholds were determined using a computer-supported linear regression break point analysis software (Vienna CPX-Tool, Austria), applying two defined regions of interest (ROI). For the detection of the first ventilatory threshold (VT_1_), a multilinear regression analysis was performed between the start of the exercise and 66% of v_max_. The ROI for the second ventilatory threshold (VT_2_) was determined between VT_1_ and v_max_. VT_1_ was defined as the first increase of VE accompanied by an increase in VE/VO_2_ without an increase in VE/VCO_2_. VT_2_ was defined as the second increase in VE accompanied by an increase in both VE/VO_2_ and VE/VCO_2_ [3] (Figure 1). The first and second accelerometer thresholds (ACT_1_ and ACT_2_) were determined from the |v| values, using the same ROIs as those used to determine the physiological thresholds. Both accelerometer thresholds were determined for the left (ACT_1L_ and ACT_2L_) and right (ACT_1R_ and ACT_2R_) leg.

### 2.5. Statistical Analysis

Data analysis and graphical illustrations were performed using GraphPad Prism 7 (GraphPad Software, San Diego, CA, USA). For conformation of normality, data were checked by the Shapiro–Wilk test. To assess differences between ventilatory thresholds and accelerometer thresholds, ANOVA and independent *t* tests were used for normally distributed data. Furthermore, the spearman correlation coefficient was calculated to evaluate the relationship between thresholds. The effect size was calculated as Cohen’s d for the comparison of running speeds at the thresholds either determined form ventilatory or accelerometer measures [31]. All data are presented as means ± standard deviation, and statistical significance was set at *p* < 0.05.

## 3. Results

Participants reached a v_max_ of 15.3 ± 1.7 km/h in the maximal incremental running test and a corresponding RER of 1.14 ± 0.05, a VO_2max_ of 53.9 ± 6.2 mL/kg/min, as well as a HR_max_ of 190.3 ± 8.7 bpm. Running speed at VT_1_ and VT_2_ was determined at 7.82 ± 0.39 and 10.91 ± 0.87 km/h, which was 51.2% and 71.1% of v_max_, respectively. Mean values of oxygen uptake, heart rate, and ventilation at VT_1_/VT_2_ as well as ACT_1_/ACT_2_ are presented in Table 1.

Accelerometer data from the tibia presented a three-phase increase with increasing speed from start to v_max_. The determination of a first and second accelerometer threshold was possible in all subjects from accelerations measured at the left or right tibia. From a total of 33 incremental running tests, 5 accelerometer data sets from the right leg had to be excluded due to invalid data recordings. 

Running speed at ACT_1_ was 7.71 ± 0.35 km/h from the left and 7.79 ± 0.33 km/h from the right leg, corresponding to 50.5% and 49.9% of v_max_. ACT_2_ was determined at a speed of 10.62 ± 0.72 km/h (69.3% v_max_) from the left and 10.74 ± 0.77 km/h (68.8% v_max_) from the right leg (Figure 2). Accelerometer thresholds of the right and left leg demonstrated a strong relationship (r = 0.86; *p* < 0.0001); however, running speeds were significantly different (*p* < 0.0019).

Comparison of physiological and accelerometer thresholds revealed slightly but significantly lower running speed at ACT_1_ and ACT_2_ compared to VT_1_ and VT_2_. There was no or just a small effect when thresholds were determined from accelerometer data compared to ventilatory parameters (Table 2, Figure 3). Additionally, running speed at VT_1_ and VT_2_ and the first and second accelerometer thresholds for both legs were significantly correlated (r = 0.98; *p* < 0.001) (Figure 4). Oxygen uptake at ACT_1_, as well as HR at ACT_2_ (for left and right), and ventilation at ACT_1R_ were found significantly different compared to their physiological reference values (Table 1). The mean differences between physiological and accelerometer thresholds were rather small and were found between 0.09–0.28 km/h for running speed, at 0.06 and 0.07 L/min for VO_2_, and at 4 bpm for HR.

Running speeds at the first and second thresholds were significantly higher in male compared to female participants. Comparison of thresholds revealed no statistical differences in the female participants but significantly lower speeds at ACT_1_ and ACT_2_ compared to VT_1_ and VT_2_ in the male participants (Table 2).

## 4. Discussion

This study shows that accelerometer data measured on the distal part of the tibia during a maximal incremental running test allow the determination of a first and a second accelerometer threshold comparable to the first and second ventilatory thresholds. Contrary to our hypothesis, both ACTs were significantly different to the ventilatory thresholds; however, the differences ranging between 0.1 and 0.3 km/h were small in that it may be suggested negligible because it is not practically meaningful. Running speed at the ventilatory and accelerometer thresholds were significantly related supporting our hypothesis of two detectable thresholds from accelerometer measures with sufficient accuracy.

Both accelerometer and ventilatory thresholds were uniformly detected within two defined regions of interest by applying a computer-supported linear regression break point analysis as shown earlier [32]. The acceleration of the left and right tibia expressed in |v| increased in a comparable manner to the increase in ventilation. With the start of running, acceleration increased steadily by a relatively low value. At a certain speed, acceleration started to increase, stronger and proportional to every 0.5 km/h increase in speed, causing a steeper time course of acceleration measures. With a further increase in intensity, at some point, accelerations once more started to increase stronger up to maximum running speed.

From the perspective of running kinematics, velocity equals the product of cadence and step length. The relation of each of these components was shown to differ depending on the running speed. At slower velocities, speed is modulated primarily by adjusting step length, whereas, at faster velocities, speed is modulated more by changes in cadence [33]. At velocities close to the maximum (sprint running), step length only shows a small increase, and velocity is primarily modulated by increases in step frequency [34,35]. An actual study by Goto et al. [34] even used the flattening of the step length at high velocities to determine an inflection point of the stride pattern. Subjects performed a series of runs at different velocities, ranging from the slowest running speed of around 8 km/h to the fastest between 29–36 km/h. About 30 trials with a distance of 20 m and a 10–30 m acceleration and deceleration zone were performed in random order. In this study, all participants tended to increase step length predominately between the slowest running speed and the determined inflection point (around 18–21 km/h) and increased cadence above the inflection point until they reached their maximal speed. However, this study refers to the sprint type of running, and results are hardly comparable to our study, although similar patterns might also occur in such an approach. The slowest and fastest running speeds in our study were much lower compared to the sprint type study from Goto et al. [34], as well as other running studies prescribing kinematics, such as the relation between speed, step length, and frequency [33,35]. However, Bailey et al. [23] also described the relationship between step length and velocity with an increase at lower speed and plateau formation at high speeds even in running speeds up to a maximum velocity of about 16 km/h, which is comparable to the maximum velocity in our study. Modulation of step length and frequency therefore seems to be the most plausible explanation for changes in the acceleration measures, respectively the |v| values during progressively increasing running speed in an incremental running test.

In our study, the first threshold was detected at a mean speed of 7.8 km/h, which corresponds to the minimum running speed in in the study from Goto et al. [34]. This was argued to be the slowest possible running speed. Running velocities below this speed were assumed to require a slower cadence, which would require “hopping” rather than running or shorter step lengths, making running similar to “jogging in place”. Velocity changes above this speed mainly occurred via changes in step length. Therefore the first inflection in accelerations in our study could be due to the transition from “unnatural slow running” to “normal running”, which was shown to start at a speed of approximately 8 km/h in a group of sedentary subjects to sprinters [34]. This is in line with Sheerin et al. [21] who suggested that the relationship of running speed and acceleration is non-linear in the transition from slow to faster speeds. This needs to be proven, if subjects with a speed significantly higher than 8 km/h at the first threshold show a different time course of acceleration measures. The second phase of linearly increasing acceleration might be explained by constant changes in the step length with increasing velocity. This phase was followed by a second inflection of acceleration, which might accrue due to further increases in frequency with a more or less consistent step length. However, an increase in acceleration expressed by the three-dimensional sum vector cannot generally be interpreted as an increase in frequency due to a faster swing of the leg and subsequent higher accelerations. A possible explanation for higher acceleration measures at higher speeds are higher ground reaction forces rather than a faster swing. Weyand et al. [36] showed that the ground force was 1.26 times greater for a runner with a top speed of 11.1 vs. 6.2 m/s, but the time taken to swing the limb into position for the next step did not vary. Anyhow, our data showed a three-phase behavior allowing the determination of three phases and two thresholds comparable to standards ventilatory thresholds. Additional biomechanical measures to detect stride kinematics as well as including subjects with higher running performance will be necessary to underpin the findings more generally.

Common accelerometer measures are used to determine energy expenditure, number of steps, as well as the intensity of exercise/activity [37,38,39]. In order to prescribe intensity, so-called accelerometer cut-points are widely used [37]. These cut-points are based on absolute intensity domains and are therefore independent of individual performance capacity. A common classification, by multiples of the metabolic equivalent (MET), is light (1.5–2.99 METs), moderate (3–5.99 METs), or vigorous (>6 METs) intensity [40]. However, at an individual level, absolute cut-points are likely to under- or overestimate the intensity compared to physiological thresholds [37,41], and it was already shown that these absolute MET-derived accelerometer thresholds just poorly estimate the individual intensity of physical activity [42,43]. The determination of two “metabolic” thresholds from accelerometer data determined from a simple incremental running test allows to overcome these shortcomings and to differentiate between three intensity zones with specific metabolic conditions.

Our study had some limitations, such as a rather homogeneous study group of trained highly active male and female individuals but a smaller number of female participants. Additional studies in inactive or sedentary subjects as well as highly trained runners are necessary to evaluate the relationship in diverse groups in order to generate general information. Additionally, we did not perform biomechanical measures, although running kinematics may be a key to understanding the patterns found. Therefore, these measures need to be performed in further studies to underpin the assumption of regular changes in stride patterns. Additionally, variable accelerometers positions (e.g., shoes) need to be proven for applicability, and a test re-test design should be applied to show reliability of this method.

## 5. Conclusions

In conclusion, our data showed that it is possible to determine two accelerometer thresholds from tibia placement during a maximal incremental running test comparable to standard ventilatory thresholds. Exercise prescription via tibia-derived accelerometer data is therefore possible, and results are promising at least for the group of trained male and female subjects tested. Accelerometers may be an easy and cheap method to prescribe exercise intensity without physiological measures.

## Figures and Tables

**Figure 1 sports-11-00171-f001:**
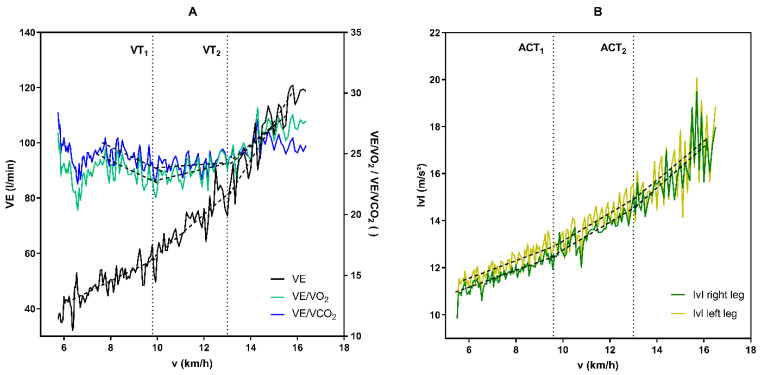
Exemplary illustration of the determination of physiological and accelerometer thresholds from a maximal incremental running test in a single subject. (**A**) Determination of the first (VT_1_) and the second (VT_2_) threshold from ventilatory variables (VE—ventilation; VE/VO_2_—equivalent for oxygen uptake; VE/VCO_2_—equivalent for carbon dioxide output) (**B**) Determination of the first and second accelerometer threshold (ACT_1_ and ACT_2_) from the acceleration measures from the left and right tibia expressed as vector magnitude (|v| = √(ax^2^ + ay^2^ + az^2^)).

**Figure 2 sports-11-00171-f002:**
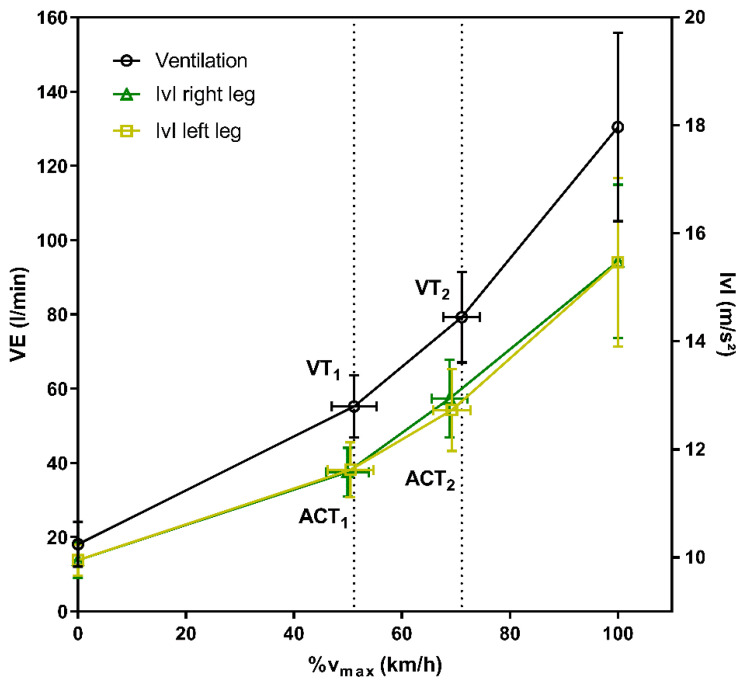
Mean values at rest, the first and second threshold and maximum of ventilation (VE), as well as accelerometer data of the left and right foot expressed as vector magnitudes (|v|).

**Figure 3 sports-11-00171-f003:**
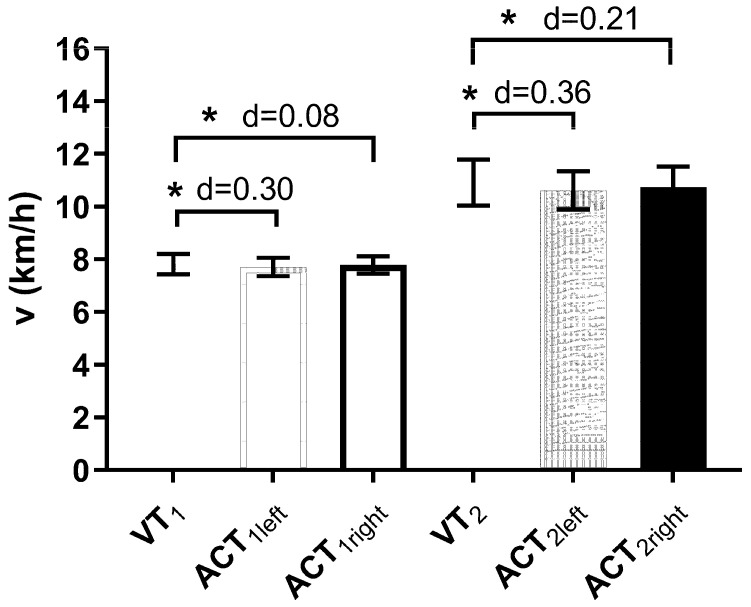
Comparison of running speeds at the first and second ventilatory (VT_1_/VT_2_) and accelerometer (ACT_1_/ACT_2_) thresholds for the left and right leg. d denotes the effect sisze and * denotes significant difference compared to ventilatory thresholds.

**Figure 4 sports-11-00171-f004:**
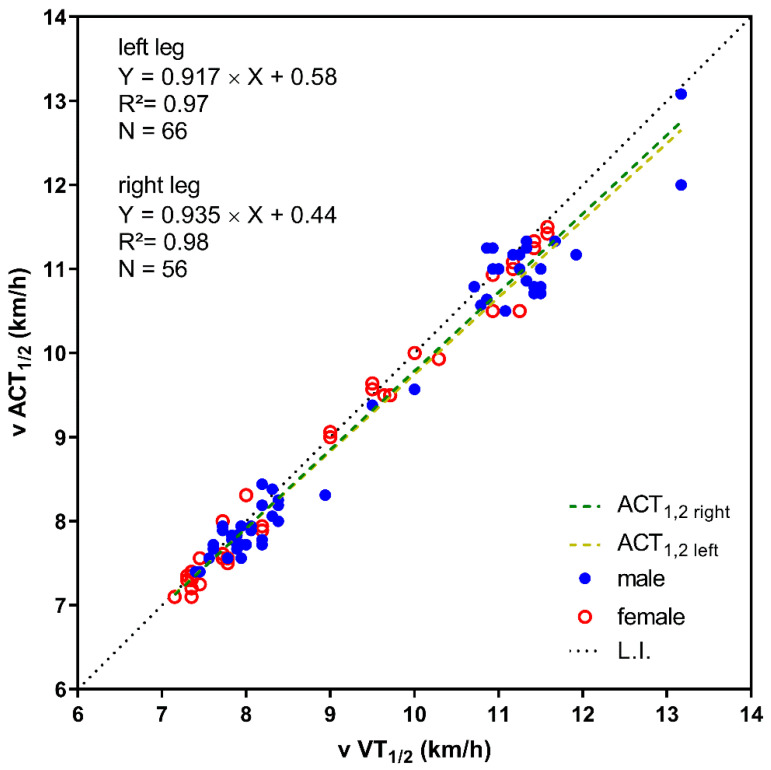
Correlation between the speed at both ventilatory thresholds (v VT_1/2_) and both accelerometer thresholds (ACT_1/2_) of the left and right leg. Male and female subjects are shown with different symbols.

**Table 1 sports-11-00171-t001:** Participant’s oxygen uptake (VO_2_), heart rate (HR), and ventilation (VE) at the first and second ventilatory and accelerometer thresholds.

	Ventilatory Thresholds		Accelerometer Thresholds			
Variables	VT_1_	VT_2_	ACT_1L_	ACT_1R_	ACT_2L_	ACT_2R_
VO_2_ (L/min)	2.06 ±0.28	2.89 ±0.5	2.13 * ±0.3	2.15 * ±0.3	2.86 ±0.5	2.87 ±0.5
Heart rate (bpm)	143.4 ±14.3	170.8 ±10.0	141.4 ±14.0	141.6 ±14.6	167.0 * ±10.9	166.8 * ±11.7
VE (L/min)	55.2 ±8.4	79.24 ±12.2	56.9 ±8.5	58.6 * ±9.2	79.1 ±11.7	79.5 ±13.2

* Significantly different to ventilatory thresholds (*p* ≤ 0.05).

**Table 2 sports-11-00171-t002:** Mean running speed at the first (T_1_) and second (T_2_) ventilatory (VT) or accelerometer (ACT) thresholds.

	Threshold	VT (km/h)	ACT Left Leg (km/h)	ACT Right Leg (km/h)
all	T_1_	7.82 ± 0.39	7.71 ± 0.35 *	7.79 ± 0.33 *
	T_2_	10.9 ± 0.9	10.6 ± 0.7 *	10.7 ± 0.8 *
female	T_1_	7.5 ± 0.25	7.45 ± 0.31	7.52 ± 0.26
	T_2_	10.12 ± 0.71	9.92 ± 0.57	10.09 ± 0.63
male	T_1_	8.01 ± 0.33	7.86 ± 0.29 *	7.91 ± 0.28 *
	T_2_	11.35 ± 0.58	11.02 ± 0.44 *	11.05 ± 0.62 *

* Significantly different to ventilatory thresholds (VT) (*p* ≤ 0.05).

## Data Availability

The data presented in this study are available on request from the corresponding author. The data are not publicly available due to hospital confidentiality.

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
