# Peer review of "Accelerometer-Derived Intensity Thresholds Are Equivalent to Standard Ventilatory Thresholds in Incremental Running Exercise"

_sports, 2023, doi:10.3390/sports11090171_

Round 1

Reviewer 1 Report

A very interesting study, with great potential for practical use. Congratulations to the Authors for the idea and execution.
Minor comments:
1. Please add how speed was controlled during the graded test.
2. I suggest adding some data confirming the achievement of VO2max (VO2max vs. VO2peak): RER, etc., and add to the methodology the criteria for achieving VO2max
3. You only used VE/VO2 and VE/VCO2 to determine thresholds - why didn't you use FEO2% and FECO2% for confirmation? see: https://doi.org/10.1159/000194758

4. Was a re-test done to determine method repeatability and validation?
5. Why did you use a treadmill in the stadium instead of a mechanical treadmill? This would allow you to control some of the interfering variables, such as environmental conditions.

6. Was the sample size calculated?

Author Response

Thank you for revising our manuscript! And sorry for the late reply, but as some co-authors were on vacation, it was not possible for us to reply earlier. Please find attached our response to your valuable comments.

Comment 1.
Please add how speed was controlled during the graded test.

Reply 1.
Speed was controlled by audio signals. We included more detailed information in the method section (line 111-116).

Comment 2.
I suggest adding some data confirming the achievement of VO2max (VO2max vs. VO2peak): RER, etc., and add to the methodology the criteria for achieving VO2max

Reply 2.
Thank you for this comment. We added some information of the determination and definition of VO2max in the method section as well as the value of the mean RER at vmax in the result section. (Line 117 - 120, Line 169)

Comment 3.
You only used VE/VO2 and VE/VCO2 to determine thresholds - why didn't you use FEO2% and FECO2% for confirmation? see: https://doi.org/10.1159/000194758

Reply 3.
As the detection of the first and second threshold by VE/VO2 and VE/VCO2 is a standard method we think it is not necessary to confirm the thresholds using a further method. However, we anyway determined the thresholds by the detection of nonlinear changes in VE accompanied by changes in VE/VO2 and VE/VCO2 (see Line 141 – 144). Additionally, we also analyzed changes in PET O2 and PET CO2, which is a similar parameter to FEO2% and FECO2%, for conformation of threshold (but these results are not shown as we think it is not necessarily due to, we already used a well-accepted standard method). Thank you for the interesting literature suggestion. 

Comment 4.
Was a re-test done to determine method repeatability and validation?

Reply 4.
No, there was no retest done. Thank you for this comment, we included this in the limitations section (line 313 and 314). As both legs showed the same pattern with no differences we suggest repeatability of data although no classis test re-test design has been performed.

Comment 5.
Why did you use a treadmill in the stadium instead of a mechanical treadmill? This would allow you to control some of the interfering variables, such as environmental conditions.

Reply 5.
We performed the tests on an outdoor 400 meter running track. Weather and environmental conditions during all tests were similar, as all tests were performed within a four weeks’ time frame. Using a mechanical treadmill would have been no option, as the running kinetics on a motorized treadmill are different compared to overground running (https://pubmed.ncbi.nlm.nih.gov/29170707/).

Comment 6.
Was the sample size calculated?

Reply 6.
No, there was no sample size calculation.

Reviewer 2 Report

The study by Schützenhöfer et al is interesting to read and presents a promising method for simplifying the determination of running intensity thresholds based solely on accelerometer data. This method of threshold detection, if confirmed, would be of great practical use for athletes and coaches for the adjustment of training loads with field tests and on a regular basis. However, there are several uncertainties regarding the presentation of the work and the methodology employed that need to be addressed before considering its acceptance for publication.

The rationale justifying the relationship between metabolic and biomechanical changes due to fatigue should be more emphasized in the introduction. Only one study by Lachlan et al (2023) covers this purpose. Other studies are cited to express that accelerometry is able to discern between soccer tasks or running intensities, but this it not the issue here. Beyond the use of accelerometers in the shin guards (i.e., tibia), there is no clear relationship between those studies and yours. It is obvious that an increase in running speed modifies stride length or frequency (in fact, they are the only way to achieve this). If necessary, this paragraph on the application of accelerometers in other sports should be briefly introduced before clearly justifying the ability of these devices to identify fatigue through changes in peak tibial acceleration. Overall, a more in-depth review of the determination of intensity thresholds through kinematic variables would fit better in this section.

Line 158: You may not include opinions or expresions like "only slightly" in the presentation of the results.

Figure 2: Please change the color of right/left line as it is difficult to distinguish between them.

Remove the hip accelerometer reference as you do not include so in the methods (or describe it in methods and be consistent in its use).

Additionally, you may remove that information regarding the accelerometer placed in the hips in the conclussion section.

Figure 4. I believe it would be better to divide the information in two graphs, one for VT1 and another for VT2.

Why n=56 for the right tibia accelerometer? Did you have to filter the data and remove outliers so that the inflection point in the accelerometry values correlated with the ventilatory thresholds?

Table 2: Data on female (as explain in lines 178-182) are not included. What does "leg" category means?"

Line 182: Again, you are stating opinions in the results section. Please remove "no practically meaninfull" expression.

Since there are significant differences between the physiological thresholds and those calculated by accelerometry, I believe that an effect size calculation could help in the interpretation of the results.

Certain expressions regarding the "validity" of this new approach should be softened as there were indeed differences between thresholds.

Could the use of accelerometry on both tibias be employed to determine the reliability or coefficient of variation of this variable?

Studies on sprint speed are included in the discussion when the speeds are not at all comparable to that of the current study. In addition, and explanations on kinematic running parameters (i.e., stride length and cadence) are also includen even though spatio-temporal parameters were not controlled.

As the authors acknowlege, VT1 speed is too low. Almost in the change from walking to running. Also VT2 did not correspond to trained young people values. Can you explain this?

MInor comments:

Line 53: please remove the word "So,"or move the comma to the correct place (after "among others,")

Line 66: A more concise language can be use to express that determining the intensity of an effort allows the quantification of time in a certain intensity zone and that this has practical applications to training.

Line 91: Change participates to participants

Line 193: remove the comma

Some considerations are specified as minor comments

Author Response

Thank you for revising our manuscript! And sorry for the late reply, but as some co-authors were on vacation, it was not possible for us to reply earlier. Please find attached our response to your valuable comments.

Comment 1:
The rationale justifying the relationship between metabolic and biomechanical changes due to fatigue should be more emphasized in the introduction. Only one study by Lachlan et al (2023) covers this purpose. Other studies are cited to express that accelerometry is able to discern between soccer tasks or running intensities, but this it not the issue here. Beyond the use of accelerometers in the shin guards (i.e., tibia), there is no clear relationship between those studies and yours. It is obvious that an increase in running speed modifies stride length or frequency (in fact, they are the only way to achieve this). If necessary, this paragraph on the application of accelerometers in other sports should be briefly introduced before clearly justifying the ability of these devices to identify fatigue through changes in peak tibial acceleration. Overall, a more in-depth review of the determination of intensity thresholds through kinematic variables would fit better in this section.

Reply 1:
Thank you for this comment. We revised this paragraph according to your suggestions and included more specific literature of tibia acceleration and the relation to running speeds (Line 51 – 82). Additionally, we made some minor changes in the introduction, aiming for better readability.

Comment 2:
Line 158: You may not include opinions or expressions like "only slightly" in the presentation of the results.

Reply 2:
We removed this term. (Line 187)

Comment 3:
Figure 2: Please change the color of right/left line as it is difficult to distinguish between them.

Reply 3:
Thank you. We changed this throughout the manuscript.

Comment 3:
Remove the hip accelerometer reference as you do not include so in the methods (or describe it in methods and be consistent in its use).

Reply 3:
Thank you for reading the manuscript carefully! We have removed all references for measurement at the hip from the manuscript (also in the conclusions).

Additionally, you may remove that information regarding the accelerometer placed in the hips in the conclussion section.

Comment 4:
Figure 4. I believe it would be better to divide the information in two graphs, one for VT1 and another for VT2.

Reply 4:
As correlation results are very similar if graphs are divided, we would like to keep the information in one graph as this would not add more relevant information to answer the research question. To increase readability of the graph we focused on the essential parameters and therefore removed the error lines of the correlation analysis in the actual graph.

Comment 5:
Why n=56 for the right tibia accelerometer? Did you have to filter the data and remove outliers so that the inflection point in the accelerometry values correlated with the ventilatory thresholds?

Reply 5:
No, we did not filter the data, as mentioned in the method section, we had to exclude 5 data sets of the right leg due to the fact that these measurements showed no plausible data in terms of raw accelerations, possibly because of initialization problems of one device. 

Comment 6:
Table 2: Data on female (as explain in lines 178-182) are not included. What does "leg" category means?"

Reply 6:
We are very sorry for this formatting mistake! Table 2 shows the numbers for female and male participants.

Comment 7:
Line 182: Again, you are stating opinions in the results section. Please remove "no practically meaninfull" expression.

Reply 7:
We removed this term from the result section.

Comment 8:
Since there are significant differences between the physiological thresholds and those calculated by accelerometry, I believe that an effect size calculation could help in the interpretation of the results.

Reply 8:
Thank you for this comment. We calculated Cohen´s d for the differences in speed between groups at the thresholds and included this in Figure 3 as well as in the methods and results section. (Line 163 – 165, Line 195 – 197)

Comment 9:
Certain expressions regarding the "validity" of this new approach should be softened as there were indeed differences between thresholds.

Reply 9:
We removed this statement from the conclusion and changed it into “results are promising…”. (Line 319)

Comment 10:
Could the use of accelerometry on both tibias be employed to determine the reliability or coefficient of variation of this variable?

Reply 10:
As measures of both legs were more or less identical, we assume a high reliability although no classic test re-test design has been performed. We mentioned that in the limits of the study section (Line 313 and 314).

Comment 11:
Studies on sprint speed are included in the discussion when the speeds are not at all comparable to that of the current study. In addition, and explanations on kinematic running parameters (i.e., stride length and cadence) are also included even though spatio-temporal parameters were not controlled.

Reply 11:
To our knowledge, there is no specific literature on acceleration during incremental running exercise. In order to explain the non-linear increase in acceleration we refer to this literature. We included a sentence so the reader can better understand this information (Line 254-256). Regarding the kinematics, we unfortunately did not measure or controlled for differences in spatio-temporal parameters although characteristic changes in stride length and cadence might be the most plausible explanation for a non-linear increase in accelerations. In further investigations, analysis of spatio-temporal parameters needs to be included. We already included this in the limitations of our study.

Comment 12:
As the authors acknowlege, VT1 speed is too low. Almost in the change from walking to running. Also VT2 did not correspond to trained young people values. Can you explain this?

Reply 12:
We agree that the speed at thresholds is rather low, possibly by the fact that although we included “trained” subjects they were not specifically trained for running rather than more resistance type subjects. On the other hand, %HRmax at VT1 (75 %HRmax) and VT2 (90 %HRmax) are comparable to the literature (Vainshelboim et al., 2020; doi:10.1016/j.chest.2019.11.022 showed 75 % and 91 % at the first and second threshold). %VO2max in our data set however, shows markedly lower values (53 % and 76 % at VT1 and 2) compared to literature (Vucetic et al., 2014 – showing a percentage of 63 % and 86 %VO2max at the first and second threshold) maybe due to the rather fast protocol with an increase in speed every 100 meters. However, as the intention was to compare thresholds within a given protocol, this is not substantial for our current study, although it may be relevant for other research questions.

(Vucetic, V., Sentija, D., Sporsi, G., Trajkovic, N., and Milanovic, Z. (2014). Comparison of ventilation threshold and heart rate deflection point in fast and standard treadmill test protocols. Acta Clin. Croat. 53, 190–203.)

Minor comments:

Comment 13:
Line 53: please remove the word "So,"or move the comma to the correct place (after "among others,")

Reply 13:
Thank you, we revised this paragraph.

Comment 14:
Line 66: A more concise language can be used to express that determining the intensity of an effort allows the quantification of time in a certain intensity zone and that this has practical applications to training.

Reply 14:
Thank you for this comment, we rephrased this sentence (Line 85 – 87).

Comment 15:
Line 91: Change participates to participants

Reply 15: Thank you

Comment 16:
Line 193: remove the comma

Reply 16: Thank you

Round 2

Reviewer 2 Report

I want to thank the authors for the work done. They have improved the original text so I consider it is now suitable for publication.